# Diagnosis of Fabry Disease Using Alpha-Galactosidase A Activity or LysoGb3 in Blood Fails to Identify Up to Two Thirds of Female Patients

**DOI:** 10.3390/ijms25105158

**Published:** 2024-05-09

**Authors:** Giovanni Duro, Monia Anania, Carmela Zizzo, Daniele Francofonte, Irene Giacalone, Annalisa D’Errico, Emanuela Maria Marsana, Paolo Colomba

**Affiliations:** Institute for Biomedical Research and Innovation (IRIB), National Research Council (CNR), 90146 Palermo, Italy; giovanni.duro@irib.cnr.it (G.D.); monia.anania@irib.cnr.it (M.A.); carmela.zizzo@irib.cnr.it (C.Z.); daniele.francofonte@irib.cnr.it (D.F.); irene.giacalone@irib.cnr.it (I.G.); annalisa.derrico@irib.cnr.it (A.D.); emanuelamaria.marsana@irib.cnr.it (E.M.M.)

**Keywords:** Fabry disease, GLA gene, α-galactosidase A, LysoGb3, lyonization

## Abstract

Anderson–Fabry disease is a lysosomal storage disorder caused by mutations in the GLA gene, which encodes the enzyme α-galactosidase A. The GLA gene is located on the X-chromosome, causing an X-linked pathology: due to lyonization, female patients usually manifest a variable symptomatology, ranging from asymptomatic to severe phenotypes. The confirmation of the clinical diagnosis of Fabry disease, achieved by measuring α-galactosidase A activity, which is usually the first test used, shows differences between male and female patients. This assay is reliable in male patients with causative mutations in the GLA gene, in whom the enzymatic activity is lower than normal values; on the other hand, in female Fabry patients, the enzymatic activity is extremely variable between normal and pathological values. These fluctuations are also found in female patients’ blood levels of globotriaosylsphingosine (LysoGb3) for the same reason. In this paper, we present a retrospective study conducted in our laboratories on 827 Fabry patients with causative mutations in the GLA gene. Our results show that 100% of male patients had α-galactosidase A activity below the reference value, while more than 70% of female patients had normal values. It can also be observed that almost half of the female patients with pathogenic mutations in the GLA gene showed normal values of LysoGb3 in blood. Furthermore, in women, blood LysoGb3 values can vary over time, as we show in a clinical case presented in this paper. Both these tests could lead to missed diagnoses of Fabry disease in female patients, so the analysis of the GLA gene represents the main diagnostic test for Fabry disease in women to date.

## 1. Introduction

Anderson–Fabry disease (Fabry disease, FD) (OMIM #301500) is a lysosomal storage disorder caused by an enzyme deficiency of genetic origin that is clinically heterogeneous, progressive, and multisystemic [1]. To date, about 50 diseases are known to be caused by deficiencies in or malfunctions of lysosomal enzymes. A reduced amount or complete absence of these enzymes leads to the storage of substrates that are not degraded within lysosomes, causing cellular and tissue damage [2]. In Fabry disease, a functional deficiency of the lysosomal hydrolase α-galactosidase A (α-GalA) leads to impaired degradation of its substrates, predominantly globotriaosylceramide (Gb3), which consequently accumulate in the lysosomes of numerous cell types [3]. The degradation of other glycosphingolipids is also impaired, but their contribution to disease development is not well known [4]. The accumulation of Gb3, with characteristic lysosomal lamelliform inclusions, can be found in several cell types, including vascular cells, endothelial cells, cardiomyocytes, podocytes, and neurons in the ganglia and in the central nervous system [5]. In addition to Gb3, LysoGb3 (globotriaosylsphingosine), a deacylated form of Gb3 that can be detected in blood circulation, has been identified [6]. The measurement of LysoGb3 in blood is used to identify patients with FD, resulting in a reliable diagnostic marker [7]. A reduction in Gb3 and LysoGb3 is often observed with enzyme replacement therapy (ERT). Gb3 and LysoGb3 are also used in patient follow-up and to evaluate the efficacy of therapy [8].

The GLA gene, encoding the α-GalA enzyme, is located on the long arm of the X-chromosome, causing an X-linked pathology [9]. In hemizygote male patients, Fabry disease can manifest a symptomatology in childhood or adolescence, with signs and symptoms that change according to age, with systemic involvement that proceeds in stages [10]. The manifestations characteristic of the classic variant of FD are neuropathic pain, angiokeratomas, cornea verticillata, hypohidrosis or anhidrosis, gastrointestinal symptoms, and albuminuria. Later in life, progressive renal involvement, the onset of left-ventricular hypertrophy (LVH), and often cerebrovascular disease arise, ultimately leading to kidney damage (CKD), heart failure, or stroke [11]. Biochemical tests show that male patients are characterized by the absence of, or a significant reduction in, α-GalA enzymatic activity and a marked increase in Gb3 and LysoGb3 levels [12]. In addition to the classic manifestation, a number of individuals with a milder symptomatology, the involvement of a single organ, and a late onset have been identified [13]. The classic clinical features are mostly absent in these cases, and residual enzyme activity is usually present [14,15].

Due to the X-linked nature of the disease, female patients show a different clinical course to male patients, which generally has less evident symptoms and is therefore more difficult to identify [16,17]. This is caused by lyonization, the inactivation of one of the two X-chromosomes, a phenomenon first described in 1961 by Mary Lyon [18]. Lyonization is a normal biological process affecting all female mammals: during embryonic development, one of the two X-chromosomes in each somatic cell becomes transcriptionally inactive, but this inactivation does not involve the same chromosome in all cells of the organism [19]. In the cells of a certain organ, the X-chromosome of paternal origin may be activated, whereas in the cells of another organ, the one of maternal origin may be activated. Because of this phenomenon, women with Fabry disease, usually heterozygotes, are a mosaic of normal and altered cells. The inactivation of the X-chromosome could affect 50% of paternally derived X-chromosomes and 50% of maternally derived X-chromosomes; however, the inactivation of the X-chromosome can sometimes occur in an unbalanced way, therefore affecting a greater percentage of X-chromosomes of paternal or maternal origin [20,21]. In Fabry disease, the severity of the symptomatology will depend on the percentage of cells in which the X-chromosome carrying the mutation in the GLA gene remains active [22]. Like in male patients, in women, there is a relationship between the levels of α-GalA activity and the severity of the disease. Female patients with absent or low α-GalA activity usually have severe clinical manifestations, but this does not mean that patients with normal enzymatic activity have no symptoms or that symptoms will not arise in the future [23]. Indeed, females carrying a mutation in the GLA gene could develop localized organ damage over time, in the heart and CNS in particular, or multisystem damage, with even more severe clinical manifestations. Until a few years ago, it was believed that women were just carriers, but today, it is believed that heterozygotes can present cardio- and cerebrovascular disorders of the same gravity as hemizygotes [24].

Lyonization also implies that the study of enzyme activity and the accumulation of LysoGb3 in blood in female patients is often unreliable. Although this is widely known, many laboratories in the world still use the study of enzymatic alterations to α-GalA and/or the dosage of LysoGb3 in blood to confirm the clinical diagnosis of Fabry disease and for newborn screening in both males and females.

In this paper, we report the results for α-GalA activity and the accumulation of LysoGb3 in the blood of 827 patients (374 males and 453 females) who presented pathogenic mutations in the GLA gene. Furthermore, we report the clinical case of a woman with an exonic mutation responsible for the classic phenotype associated with the severe form of the disease. At the time of the study, the patient had normal enzyme activity, no accumulation of LysoGb3 in the blood, and no symptoms. After a few years, however, the patient suffered a stroke, proteinuria, and kidney damage, with the presence of accumulated blood LysoGb3 increasing over the time.

## 2. Results

In the last 18 years, the research group of the Centre for Research and Diagnosis of Lysosomal Storage Disorders at IRIB-CNR (Institute for Biomedical Research and Innovation of the National Research Council) have studied over 35,000 samples from patients with signs and symptoms referable to Fabry disease (average age: 41.8 years old, 58.8% male and 41.2% female, with average ages of 42.4 and 40.2 years old, respectively). In this population, 1182 subjects with nucleotide alterations in coding regions or in regulatory regions (splicing sites) of the GLA gene were found: among these, 827 had a mutation responsible for the classic variant or late-onset variants of Fabry disease, 355 subjects had a mutation considered to be a genetic variant of uncertain significance (GVUS), and the remainder had no variations in the coding or regulatory regions of the GLA gene. In Table 1, we show the characteristics of the population in which mutations in the GLA gene responsible for Fabry disease were found, the results of the measurement of α-galactosidase A enzyme activity, and the dosage of the specific substrate (LysoGb3 in blood) in all patients, in male patients, and in female patients. We also show the values obtained in individuals with the classic and late-onset variants of the disease. The female patient population had a normal α-GalA activity (6.1 nmol/mL/h) on average, with no significant differences between the classic and late-onset variants, while the male patient population showed pathological enzymatic activity (average 0.8 nmol/mL/h), which was significantly lower in the case of the classic variant (average 0.2 nmol/mL/h) than the late-onset variant (1.3 nmol/mL/h) (Table 1). Regarding the dosage of LysoGb3 in blood, females with the classic variant of the disease had average values that were pathological (6.3 nmol/L), although these were lower than the values found in male patients (average 44.4 nmol/L), while females with a late-onset variant had average values that were normal (1.8 nmol/L) (Table 1).

The results obtained in this study show that in female patients, the manifestation of the disease is delayed compared to male patients, the activity of α-GalA enzyme is higher on average, and the amount of LysoGb3 in the blood is lower. To elaborate, our results show that in all male patients with either classic or late-onset variants of Fabry disease, the enzyme activity is lower than the reference value (Table 2), while in affected female patients, the activity of the α-GalA enzyme has pathological values in 25.9% of patients with late-onset variants and 29.9% of patients with the classic variant, with an overall average value of 28.2% (Table 2). In Figure 1A, the stripes indicate the percentage of female patients with normal enzyme activity, while dots indicate the percentage of female patients with pathologic enzyme activity. Furthermore, we observed in patients undergoing follow-up that α-GalA activity does not substantially vary over the course of a lifetime.

The blood accumulation of LysoGb3 also reflects the differences found between male and female patients. While 100% of male patients with the classic or late-onset variants of Fabry disease show a pathological accumulation of LysoGb3 in the blood (Table 3), 57.6% of female patients show pathological values, and 42.4% show normal values (Table 3). Analyzing these data in more detail, we can see that a pathological value of blood LysoGb3 was measured in 82.8% of female patients with the classic phenotype, but only in 19.4% of patients with the late-onset variant (Table 3). In Figure 1B, the stripes indicate the percentage of female patients with normal values of LysoGb3 in blood, while points indicate the percentage of subjects with pathological values of LysoGb3.

We also observed that blood LysoGb3 accumulation can vary significantly over a woman’s lifetime, ranging from normal to pathological values. We show here the monitoring of α-GalA activity and blood LysoGb3 levels in a female patient who was followed up for 3 years before starting therapy. The patient, a 26-year-old female at the time of the first study, came to our attention as the daughter of a male patient diagnosed with the c.718_719delAA mutation in the GLA gene, responsible for the classic variant of Fabry disease. At that time, the patient, who inherited the same mutation from her father, had not shown any signs or symptoms referable to Fabry disease; her α-GalA activity was 3.6 nmol/mL/h (normal values > 3 nmol/mL/h) and her blood LysoGb3 was 0.99 nmol/L (normal values < 2.3 nmol/L) (Figure 2). The following year, at age 27, her enzyme activity was 3.8 nmol/mL/h (normal values > 3 nmol/mL/h), and her blood LysoGb3 increased to 3.34 nmol/L (normal values < 2.3 nmol/L) (Figure 2). A year later, at age 28, her α-GalA activity was 3.5 nmol/mL/h (normal values > 3 nmol/mL/h), and her blood LysoGb3 further increased to 5.98 nmol/L (normal values < 2.3 nmol/L) (Figure 2). In addition to the increase in blood LysoGb3, the patient suffered a stroke, proteinuria, and kidney damage over the final two years. She was therefore started on enzyme replacement therapy, which led to a reduction in the accumulation of substrates in her blood: at age 29, her blood LysoGb3 level was 4.30 nmol/L (normal values < 2.3 nmol/L) (Figure 2).

## 3. Materials and Methods

### 3.1. Patients

Peripheral blood was collected, using EDTA as an anticoagulant, and dried on absorbent paper (dried blood spot, DBS). Genetic and enzymatic studies were performed at the Centre for Research and Diagnosis of Lysosomal Storage Disorders of IRIB-CNR in Palermo.

### 3.2. Genetic Analysis

Genomic DNA was isolated from dried blood spot using silica-coated magnetic particles in a robotic workstation designed for automated purification of nucleic acids. DNA concentrations were estimated using a biophotometer (Eppendorf, Hamburg, Germany). The search for mutations in the GLA gene was performed via Sanger sequencing. Eight pairs of primers were designed to analyze eight target regions containing the seven exons of the GLA gene, including the flanking regulatory sequences, and the cryptic exon. PCR products were purified and sequenced at Eurofins Genomics (Ebersberg, Germany).

### 3.3. α-Galactosidase A Activity Assay

α-galactosidase A activity assays were performed using the dried-blood filter paper (DBFP) test described by Chamoles et al. [25], with some modifications [unpublished data]. A spot of 10 μL of blood in a circle of paper 6 mm in diameter was placed into a 96-well plate, suitable for fluorometric assays, and incubated for 18 h at 37 °C in a thermomixer; the reaction was terminated by the addition of 250 μL of 0.1 mol/L ethylenediamine (pH 11.4). The background fluorescence, i.e., fluorescence which was not due to the specific enzyme activity, was determined for each sample, conducting another reaction in the presence of 0.14 mmol/L 1-deoxygalactonojirimycin (DGJ, the inhibitor of alpha galactosidase A) in citrate phosphate buffer (pH 4.5). This background was subtracted from the fluorescence of the sample. In each assay, we added positive and negative controls and a calibration curve with 4-methylumbelliferone. Normal values were >3.0 nmol/mL/h.

### 3.4. LysoGb3 Determination

The determination of LysoGb3 in blood was performed via tandem mass spectrometry (MS/MS) methodology, as previously described by Polo et al. [26].

## 4. Discussion

Fabry disease is a genetic disorder that is difficult to diagnose because of its peculiarities: the heterogeneity and complexity of its clinical manifestations, its non-specific signs and symptoms, and the fact that it is a rare and often little-considered disease [27]. In patients with clinical suspicion of Fabry disease, but also in newborn screenings, the first laboratory investigation carried out is usually the study of α-galactosidase A enzyme activity. This analysis is reliable in male individuals for distinguishing patients with causative mutations in the GLA gene from individuals not affected by the disease. Indeed, in male patients with Fabry disease, the enzymatic activity is always lower than the reference value, both for the classic variant of the disease and for the late-onset variants. On the other hand, in female patients, the activity of α-GalA is extremely variable, oscillating between normal and pathological values, due to lyonization. The retrospective study conducted in our laboratories on 827 patients (374 males and 453 females), all with a causative mutation in the GLA gene, showed that 100% of male patients had an α-GalA activity lower than the reference value, regardless of the phenotype. Therefore, the enzyme analysis would be able to identify all of them. Conversely, more than 70% of female patients had normal enzyme activity, both with the classic variant and with late-onset variants, which would therefore escape diagnosis if the enzymatic test alone was used. The difference found between male and female patients is not due to the methodology used for the assay, but the characteristics of lyonization. These results lead us to consider that, when the analysis of α-galactosidase A activity is carried out in a woman affected by Fabry disease, it is possible to find normal values deriving from the pool of normal cells. Therefore, measuring enzyme activity as the first diagnostic test in these subjects could be preventing correct diagnosis. Although this is widely known, ours is the first retrospective study derived from the analysis of a large number of Fabry patients (827, of whom 453 were females) over 18 years. Other works in the literature, derived from the study of fewer subjects, report lower percentages of missed diagnoses in female patients (e.g., Linthorst et al., 2005 [28]). It is also important to underline that newborn screenings carried out through the study of enzymatic alterations yield a high percentage of false negatives in female neonates, evidenced by the fact that almost all Fabry patients identified in this way are male [29,30].

The differences observed in the enzyme activity in male and female patients are also found when the accumulation of LysoGb3 in blood is determined. Our results show that while 100% of male patients have pathological values of LysoGb3 in their blood, 42.4% of females with Fabry disease have normal blood LysoGb3 values, with a significant difference between subjects with the classic variant compared to patients with late-onset variants. In the first case, 82.8% of patients have pathological values of blood LysoGb3, although these values are lower than the accumulation detected in males with the classic variant, as previously reported [12]. Therefore, using this test for diagnosis, 17.2% of patients would be missed. However, if we consider female patients with late-onset variants, only 19.4% of them have pathological values of blood LysoGb3, while 80.6% have normal values and would therefore escape diagnosis. Therefore, we can state that the measurement of LysoGb3 in blood is a more reliable test than the measurement of α-galactosidase A activity for female patients with the classic variant of Fabry disease, but it still misses a very high percentage of female patients with late-onset variants of the disease. For this reason, we want to strongly emphasize that, to date, genetic analysis represents the most effective diagnostic test for Fabry disease in women, both in the case of diagnostic confirmation and in newborn screening.

Missed diagnosis in female Fabry patients carries potential clinical consequences: above all, organ damage progresses irreversibly, and, in some cases, patients can manifest a severe symptomatology like that seen in male patients (if an unfavorable imbalance in the X-chromosomes inactivation occurs). Furthermore, missed diagnoses result in a failure to identify other affected patients through family assessments and the failure of genetic counselling, in addition to a worse quality of life [24].

Another consequence of lyonization is that the levels of LysoGb3 circulating in female patients can increase over time, contributing to the development of severe symptoms, unlike α-galactosidase A activity, which remains almost constant throughout an individual’s life span. In this paper, we report the interesting case of a young woman with an exonic mutation in the GLA gene, c.718_719delAA, responsible for the classic, severe phenotype of Fabry disease. The mutation was inherited from her father, who was diagnosed late and died at the age of 56. At the time of the study, the patient had normal α-GalA activity, normal blood LysoGb3 values, and no symptoms. The study of the patient’s enzyme activity in the following two years revealed the α-GalA activity remained almost unchanged, while the accumulation of LysoGb3 increased significantly. During this period, the patient experienced a stroke, proteinuria, and renal damage, and she is now on enzyme replacement therapy. This clinical case confirms that, in female patients, the accumulation of specific substrates can increase over time, and that typical signs and symptoms of the disease, even of a certain severity, can arise later in life, confirming the importance of a complete follow-up in women with mutations in the GLA gene.

## 5. Conclusions

Our retrospective study of 827 patients with mutations in the GLA gene responsible for Fabry disease suggests that studies of α-galactosidase A activity and blood LysoGb3 dosage in female subjects are not reliable and therefore cannot be used as the first diagnostic test for diagnostic confirmation, nor for the screening of newborns. Genetic testing is the most effective diagnostic test for Fabry disease in women.

Fabry disease has progressive effects on many organs, so it cannot be excluded that clinical manifestations of the disease may appear later in life in women. Therefore, a careful follow-up should always be planned in these patients, even in the absence of symptomatology at the time of disease diagnosis, and in the presence of normal enzyme activity and/or LysoGb3 in the blood.

## Figures and Tables

**Figure 1 ijms-25-05158-f001:**
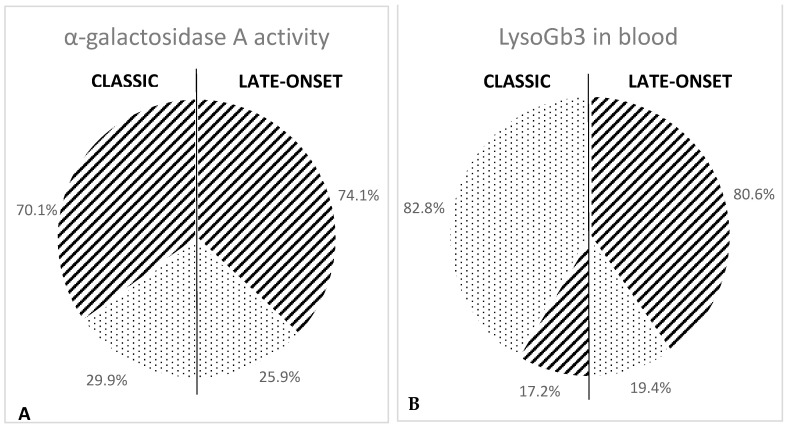
(**A**): α-galactosidase A activity in female patients with classic (on the left) or late-onset (on the right) Fabry disease. Stripes indicate the percentage of subjects with normal enzymatic activity; points indicate the percentage of subjects with pathologic enzyme activity. Enzyme activity is measured in nmol/mL/h; normal values > 3 nmol/mL/h. (**B**): Blood accumulation of LysoGb3 in female patients with classic (on the left) or late-onset (on the right) Fabry disease. Stripes indicate the percentage of subjects with normal values for LysoGb3; dots indicate the percentage of subjects with pathological values for LysoGb3. LysoGb3 is measured in nmol/L; normal values: 0.1–2.3 nmol/L.

**Figure 2 ijms-25-05158-f002:**
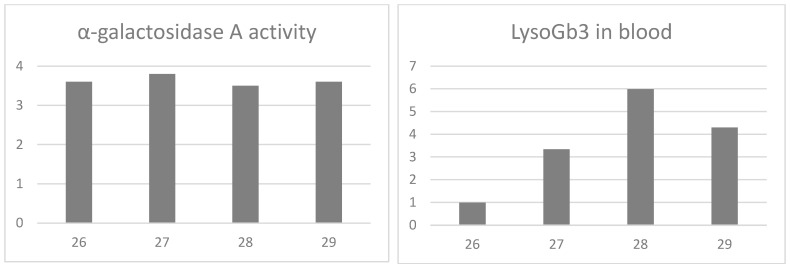
α-galactosidase A activity and LysoGb3 concentration in blood of a female patient at 26, 27, 28, and 29 years old. Enzyme activity is measured in nmol/mL/h; normal values > 3 nmol/mL/h. LysoGb3 is measured in nmol/L; normal values < 2.3 nmol/L.

**Table 1 ijms-25-05158-t001:** α-galactosidase A activity and blood accumulation of LysoGb3 in 827 subjects with an identified genetic alteration in the GLA gene (probands + relatives). Enzyme activity is measured in nmol/mL/h; normal values > 3 nmol/mL/h. LysoGb3 is measured in nmol/L; normal values 0.1–2.3 nmol/L.

Overall	Patients	Male Patients	Female Patients
Patients (%)	827	374 (45.2%)	453 (54.8%)
Average age (min-max)	41.3 (0–87)	40.5 (0–75)	41.9 (0–87)
Average α-Gal A activity	3.8	0.8	6.1
Average LysoGb3	14.2	26.2	4.5
**Classic Phenotype**	**Patients**	**Male Patients**	**Female Patients**
Patients (%)	456	188 (41.2%)	268 (58.8%)
Average age (min-max)	40.2 (0–87)	38.1 (0–63)	41.7 (0–87)
Average α-Gal A activity	3.7	0.2	6.0
Average LysoGb3	21.9	44.4	6.3
**Late-Onset**	**Patients**	**Male Patients**	**Female Patients**
Patients (%)	371	186 (50.1%)	185 (49.9%)
Average age (min-max)	42.6 (0–86)	42.9 (0–75)	42.2 (6–78)
Average α-Gal A activity	4.1	1.3	6.9
Average LysoGb3	4.0	6.3	1.8

**Table 2 ijms-25-05158-t002:** α-galactosidase A activity in 827 patients with a genetic alteration in the GLA gene identified (probands + relatives). Enzyme activity is measured in nmol/mL/h; normal values > 3 nmol/mL/h.

Overall	Patients with Mutations	Patients with Mutations and Activity < 3 nmol/mL/h	% of Patients with Mutations and Activity < 3 nmol/mL/h	Patients with Mutations and Activity > 3 nmol/mL/h	% of Patients with Mutations and Activity > 3 nmol/mL/h
Male Patients	374	374	100%	-	-
Female Patients	453	128	28.2%	325	71.8%
Classic					
Male Patients	188	188	100%	-	-
Female Patients	268	80	29.9%	188	70.1%
Late-Onset					
Male Patients	186	186	100%	-	-
Female Patients	185	48	25.9%	137	74.1%

**Table 3 ijms-25-05158-t003:** Blood accumulation of LysoGb3 in 827 patients with a genetic alteration in the GLA gene identified (probands + relatives). LysoGb3 is measured in nmol/L; normal values 0.1–2.3 nmol/L.

Overall	Patients with Mutations	Patients with Mutations and LysoGb3 > 2.3 nmol/L	% of Patients with Mutations and LysoGb3 > 2.3 nmol/L	Patients with Mutations and LysoGb3 < 2.3 nmol/L	% of Patients with Mutations and LysoGb3 < 2.3 nmol/L
Male Patients	374	374	100%	-	-
Female Patients	453	261	57.6%	192	42.4%
Classic					
Male Patients	188	188	100%	-	-
Female Patients	268	222	82.8%	46	17.2%
Late-Onset					
Male Patients	186	186	100%	-	-
Female Patients	185	36	19.4%	149	80.6%

## Data Availability

The raw data supporting the conclusions of this article will be made available by the authors on request.

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
