# Peer review of "Diagnosis of Fabry Disease Using Alpha-Galactosidase A Activity or LysoGb3 in Blood Fails to Identify Up to Two Thirds of Female Patients"

_ijms, 2024, doi:10.3390/ijms25105158_

Round 1

Reviewer 1 Report (Previous Reviewer 3)

Comments and Suggestions for Authors

I have previously reviewed this article and find it to be a crucial contribution to the field. The topic is important, and I believe it merits publication due to its potential impact on advancing our understanding in this area. The findings presented in the manuscript could have far-reaching implications for future research and clinical practice. I recommend that it be published to disseminate these valuable insights to the scientific community.

Reviewer 2 Report (Previous Reviewer 2)

Comments and Suggestions for Authors

This is a very good review of the pitfalls in the diagnosis of Fabry disease

Reviewer 3 Report (Previous Reviewer 1)

Comments and Suggestions for Authors

This retrospective study indicates that neither the alpha-Galactosidase A activity assay nor lysoGb3 in blood accurately diagnose Fabry disease in female patients. Analysis of the GLA gene is emphasized as a more appropriate diagnostic test for Fabry disease in women, providing valuable insights. I am pleased with the revised version and support its publication.

This manuscript is a resubmission of an earlier submission. The following is a list of the peer review reports and author responses from that submission.

Round 1

Reviewer 1 Report

Comments and Suggestions for Authors

Satisfied with the improvement and the revised version

Reviewer 2 Report

Comments and Suggestions for Authors

The authors report a retrospective study on 827 patients demonstrating that enzyme activity and lyso Gb3 level measured on DBS correlate well with genotype (classic vs late onset) in males - but not in females - with Fabry disease.

The study is large, rigorous and well presented. However, the data are not new - the conclusion is well known. The discussion notes the difficulty in diagnosing Fabry, which is largely due to disease heterogeneity and lack of awareness.

Comments on the Quality of English Language

Generally - too long; could be shortened and presented as a brief review

Line 77 - it needs to be clear that even within an organ individual cells in a female will demonstrate mosaicism.

Reviewer 3 Report

Comments and Suggestions for Authors

This manuscript retrospectively analyzed a large cohort of patients with Fabry disease (total: 827, male: 374, female: 453) and highlights a crucial finding: measuring α-galactosidase A activity and serum lyso-Gb3 levels may fail to identify up to two-thirds of female patients. Despite the long-standing recognition that the concept of relying on α-galactosidase A activity to identify female patients with Fabry disease is unreliable, and the fact that lyso-Gb3 levels may not always be elevated in some female Fabry patients has been reported in numerous studies, it still hasn't received significant attention from many Fabry disease diagnostic or research centers. To my knowledge, some centers continue to rely solely on enzyme activity and lyso-Gb3 levels for screening high-risk female Fabry disease patients or even for diagnosing suspected female Fabry patients.

This manuscript collected and analyzed a substantial number of female Fabry patients, confirming that over 70% of female patients with pathogenic mutations in the GLA gene exhibited normal α-galactosidase A activity values, and almost half showed normal lyso-Gb3 values in blood. The study suggests that, to date, analyzing the GLA gene remains the most reliable diagnostic test for Fabry disease in women.

I highly recommend that this manuscript be published.

Comments on the Quality of English Language

While there are no specific comments, the writing in this manuscript is somewhat disjointed and would benefit from the expertise of a native English speaker familiar with Fabry disease to refine it.